# Liraglutide Activates Glucagon-Like Peptide 1 Receptor to Attenuate Hyperglycemia through Endogenous Beta-Endorphin in Diabetic Rats

**DOI:** 10.3390/ph13110407

**Published:** 2020-11-20

**Authors:** Kai-Chun Cheng, Ying-Xiao Li, Po-Chuen Shieh, Juei-Tang Cheng, Chia-Chen Hsu

**Affiliations:** 1Department of Pharmacy, College of Pharmacy, Tajen University, Pingtung 907, Taiwan; satatonma@gmail.com (K.-C.C.); pochuen7166@yahoo.com.tw (P.-C.S.); 2Pharmacological Department of Herbal Medicine, Graduate School of Medical and Dental Sciences, Kagoshima University, Kagoshima 890-8544, Japan; 3Department of Psychosomatic Internal Medicine, Graduate School of Medical and Dental Sciences, Kagoshima University, Kagoshima 890-8544, Japan; 4Department of Nursing, Tzu Chi University of Science and Technology, Hualien City 970, Taiwan; bebeli009@hotmail.com; 5Department of Medical Research, Chi-Mei Medical Center, Tainan City 710, Taiwan; 6Graduate Institute of Gerontology and Health Care Management, Chang Gung University of Science and Technology, Taoyuan City 613, Taiwan; 7Department of Otorhinolaryngology, Taipei City Hospital, Taipei City 110, Taiwan; 8Department of Exercise and Health Sciences, University of Taipei, Taipei City 110, Taiwan

**Keywords:** liraglutide, GLP-1 receptor, beta-endorphin, adrenal gland, diabetic rats

## Abstract

Liraglutide, an acylated analog of glucagon-like peptide 1 (GLP-1), could improve glycemic control in diabetes. Moreover, endogenous opioid peptides play a role in blood sugar regulation. Since GLP-1 receptors are also expressed in extra-pancreatic tissues, this study investigates the effect of liraglutide on endogenous opioid secretion in type 1-like diabetes. The endogenous opioid level was determined by enzyme-linked immunosorbent assay. The direct effect of liraglutide on endogenous opioid secretion was determined in the isolated adrenal medulla. Acute treatment with liraglutide dose-dependently attenuated hyperglycemia, and increased the plasma opioid neuropeptide, beta-endorphin (BER) levels in diabetic rats. These effects have been blocked by GLP-1 receptor antagonist, naloxone. Additionally, the effects of liraglutide were markedly reduced in adrenalectomized diabetic rats. In the isolated adrenal medulla, liraglutide induced BER secretion and increased the BER mRNA levels. Subcellular effects of liraglutide on the adrenal gland were further identified to mediate through the exchange proteins directly activated by cAMP, mainly using the pharmacological blockade. After repeatedly administering liraglutide, metabolic changes in diabetic rats were investigated, and genes associated with gluconeogenesis in the liver were downregulated. Naloxone pretreatment inhibited these effects of liraglutide, indicating the involvement of endogenous opioids. The present study indicated that liraglutide had an acute effect of reducing hyperglycemia by regulating endogenous opioid BER and modifying the glucose homeostasis.

## 1. Introduction

Liraglutide is a clinical agent with an acylated GLP-1 analog developed using recombinant DNA technology. It shares 97% homology with endogenous GLP-1, which is better than exenatide, that shares 57% homology. Liraglutide binds to the GLP-1 receptor (GLP-1R), a G-protein-coupled receptor located on the membrane cell surface, and it induces intracellular activation by increasing the level of cyclic adenosine monophosphate (cAMP). Liraglutide influences the secretions of both pancreatic beta and alpha cells. The most important effect of liraglutide is the glucose-dependent stimulation of insulin secretion that reduces blood glucose. It also reduces glucagon secretion from pancreatic beta-cells by targeting GLP-1R, inhibiting hepatic glucose production. It has been suggested that liraglutide increases beta-cell mass and function by stimulating replication and inhibiting apoptosis [1,2]. In isolated human pancreatic islet cells, liraglutide was found to promote beta-cell proliferation and inhibit interleukin-1β-induced apoptosis after a 4 day incubation period [3]. This implies that liraglutide is also useful in treating type 2 diabetes because progressive islet dysfunction is associated with hyperglycemia. In addition, liraglutide may delay gastric emptying and acid secretion, thereby reducing post-prandial hyperglycemia and increasing satiety. Moreover, activation of GLP-1R in the brain is known to increase satiety and reduce food intake. Furthermore, liraglutide has been suggested to improve cognitive function, reduce amyloid plaque deposition, and enhance long-term synaptic transmission in animal models of dementia, indicating a possible application for preventing Alzheimer’s disease [4]. Clinically, liraglutide induces a marked loss in body weight [5], improves myocardial [6], and endothelial [7] functions, and ameliorates hyperlipidemia and hypertension. Thus, liraglutide monotherapy and/or combination therapy with other antidiabetic drugs is associated with an improvement in glycemic control [8]. Liraglutide is associated with a lower incidence of hypoglycemic events compared to other antidiabetic drugs, which has contributed to its applications in clinical practice. Owing to such pleiotropic effects, liraglutide is the drug of choice after metformin. Interestingly, it also has a protective effect against cardiovascular risk [9]. Therefore, liraglutide is widely used in treating patients with type 2 diabetes, as it is effective and well-tolerated, has a low risk of hypoglycemia and cardiovascular disorders, and induces sustained weight loss.

Recently, studies have focused on the extra-pancreatic effects of GLP-1 analogs for additional pharmacological effects, particularly those due to chronic stimulation of GLP-1R [10]. GLP-1 analogs are known to restore and/or maintain regular functions in the brain and peripheral tissues [11]. These effects of liraglutide might be mediated by mechanisms in addition to insulin-dependent actions.

Opioid peptides are endogenous hormones that primarily act as analgesics. Opiates and opioid medications have been used as therapeutic tools for a long time, and they continue to be the foundation for many cogent pain regimens [12]. Research has shown that endogenous opioid beta-endorphin (BER) inhibits glucose production in conscious animals [13]. Additionally, BER is involved in the antihyperglycemic effects of herbal extracts, including catalpol [14] and myricetin [15], in rats with type-1 diabetes [16]. Recently, catalpol [17] and myricetin [18] were shown to be GLP-1R agonists. However, there is no evidence showing that GLP-1R activation promotes BER secretion. Therefore, the present study evaluated whether GLP-1R plays a role in BER secretion. We used rats with type-1 diabetes to eliminate the mediation of endogenous insulin, as described previously [16].

## 2. Results

### 2.1. Liraglutide Attenuates Hyperglycemia with Increased Plasma BER Levels in Rats with Type-1 Diabetes

Acute treatment with liraglutide markedly reduced hyperglycemia in rats with type-1 diabetes. This time-dependent effect of liraglutide was induced 30 min after injection, and it reached a plateau at 120 min (Figure 1A). Notably, as shown in Figure 1B, liraglutide-attenuated hyperglycemia was reversed by naloxone in a dose-dependent manner. Additionally, plasma BER levels were also significantly increased by liraglutide, as shown in Figure 1C. Pretreatment with GLP-1R antagonist Ex9 [19] inhibited liraglutide-induced elevation in plasma BER (Figure 1C). Blockade of liraglutide-induced action by the antagonist was observed in a dose-dependent manner. The total blockade of liraglutide-induced action by Ex9 at 100 μg/kg IP was consistent with a previous report [20].

In addition, the plasma insulin level in rats with type-1 diabetes was significantly (*p* < 0.01) lower (2.15 ± 0.98 µU/mL; *n* = 6) than that in normal rats (12.64 ± 3.19 µU/mL; *n* = 6). However, acute treatment with liraglutide for 120 min at the highest dose reduced plasma insulin level in rats with type-1 diabetes to 1.98 ± 0.72 µU/mL (*n* = 6), which was not significantly different (*p* > 0.05) from that in vehicle-treated rats (2.21 ± 0.84 µU/mL; *n* = 6). Therefore, liraglutide did not affect the insulin level in rats with type-1 diabetes.

### 2.2. Liraglutide Induced BER Secretion from Adrenal Gland in Rats with Type-1 Diabetes

Bilateral adrenalectomy was performed in rats, as described previously [21]. T The control group received a sham surgery. A liraglutide-induced anti-hyperglycemic effect (Figure 2A) and an increase in plasma BER level (Figure 2B) were both absent in adrenalectomized rats, unlike in control rats.

### 2.3. Direct Effect of Liraglutide on The Adrenal Medulla to Induce BER Secretion In Vitro

Next, the adrenal medulla isolated from streptozotocin (STZ)-treated rats was used to study the direct effect of liraglutide after 1 h incubation in vitro, as described in our previous report [21]. Liraglutide enhanced BER secretion from the adrenal medulla (Figure 3A), and this action was inhibited by Ex9 at concentrations effective to block GLP-1R [19]. Additionally, pretreatment with ESI-09, an exchange protein inhibitor directly activated by cAMP (Epac), attenuated the effect of liraglutide (Figure 3A). However, the protein kinase A (PKA) inhibitor H-89 failed to modify liraglutide’s effect. Similarly, liraglutide upregulated the BER gene in the isolated adrenal medulla, as assessed using proopiomelanocortin (POMC) mRNA levels. Results (Figure 3B) from the pharmacological blockade using pretreatment with specific inhibitors demonstrated that liraglutide induced the upregulation of POMC mRNA levels in the adrenal medulla via GLP-1R activation to enhance Epac but not PKA.

### 2.4. Liraglutide Modified Glucose Homeostasis through BER in Rats with Type-1 Diabetes

T To evaluate the effect on glucose regulation in rats with type-1 diabetes, liraglutide was repeatedly administered once daily to attenuate hyperglycemia. One week later, there was a marked reduction in fasting plasma glucose levels in treatment groups compared to those in the vehicle-treated group. Additionally, another group that received the opioid receptor antagonist naloxone injection 30 min before liraglutide was used for comparison. Their fasting plasma glucose was reversed to a level similar to that in the vehicle-treated group (Figure 4A).

Diabetic rats were sacrificed, and the liver and soleus muscle tissues were isolated to investigate changes in the factors involved in glucose homeostasis. As shown in Figure 4A, the limiting enzyme of glucose homeostasis in the liver, phosphoenolpyruvate carboxykinase (PEPCK), was markedly inhibited by liraglutide after repeated treatment. Thus, the opioid receptor blockade by naloxone may overcome PEPCK inhibition by liraglutide. The level of another liver regulatory enzyme, GSK3β, followed a similar pattern (Figure 4B). Notably, reduced expression of GLUT4 in the soleus muscle of diabetic rats was markedly upregulated by liraglutide; this was also reversed by naloxone (Figure 4C). Additionally, PI3K phosphorylation was altered in a similar manner (Figure 4C). Therefore, acute treatment with liraglutide may attenuate hyperglycemia via BER, which regulates glucose homeostasis in rats with type-1 diabetes.

## 3. Discussion

In the present study, we found that acute treatment with liraglutide attenuates hyperglycemia in rats with type-1 diabetes in an opioid-dependent manner. Additionally, acute liraglutide administration activates GLP-1R to enhance BER secretion from the adrenal gland, which has not been mentioned before.

Glucagon-like peptide 1 (GLP-1) is a gut-derived incretin hormone that improves glucose control in diabetes [22]. Previous study found that GLP-1R might also be expressed in extra-pancreatic tissues [23], which implies that this hormone also acts via unidentified mechanisms. Liraglutide (Victoza) is a GLP-1 analog that is injected once daily. It shows 97% homology to human native GLP-1 and is an intermediate-acting product that works differently from the oral product semaglutide (Ozempic) [24]. Clinically, liraglutide is a first-line agent for treating diabetes, and it is the preferred choice after metformin with a low risk of hypoglycemia and a protective effect on cardiovascular risk [9]. In rats with type-1 diabetes, liraglutide induced an anti-hyperglycemic effect, which was reversed by naloxone. This led us to study the role of endogenous opioids in the anti-hyperglycemic action of liraglutide.

In addition to reducing hyperglycemia in a dose-dependent manner, a single liraglutide bolus increased plasma BER in rats with type-1 diabetes. Pretreatment with Ex9, an antagonist of GLP-1R, reversed the liraglutide-induced increase in plasma BER levels. Endogenous BER has physiological functions that are mainly mediated by the activation of opioid µ-receptor [25]. Therefore, increased BER levels seem to be associated with the plasma glucose-lowering effect of liraglutide via the activation of GLP-1R in rats with type-1 diabetes. It has been documented that endogenous BER inhibits glucose production in conscious animals [13]. Endogenous BER or opioid receptor activation has also been demonstrated to positively regulate glucose homeostasis in type 1-like animal models [21]. Therefore, it is reasonable to deduce that liraglutide induces an anti-hyperglycemic effect via the activation of GLP-1R through an increased endogenous BER in diabetic rats with insulin deficiency.

Additionally, we investigated the source of endogenous BER following liraglutide treatment. It has been demonstrated that the adrenal gland is responsible for increased circulating BER levels in rats with type-1 diabetes [26]. Our results are consistent with these findings because the liraglutide-mediated increase in plasma BER level was reversed in adrenalectomized diabetic rats. Moreover, we isolated the adrenal medulla from diabetic rats to investigate the direct effect of liraglutide. Liraglutide enhanced BER secretion from the adrenal medulla through GLP-1R activation. Interestingly, pharmacological identification showed that Epac was the primary subcellular signal of GLP-1R responsible for this influence of liraglutide. Increased POMC mRNA levels in the adrenal medulla seemed to follow the same pathway. Therefore, we demonstrated a novel mechanism in which liraglutide activates GLP-1R to increase BER secretion in the adrenal medulla via Epac but not via PKA.

We also investigated changes in glucose homeostasis in rats with type-1 diabetes after repeat treatment with liraglutide for seven days. In addition to reducing hyperglycemia, particularly the fasting plasma glucose level, the higher expression of the hepatic enzyme PEPCK involved in gluconeogenesis was markedly reduced by liraglutide. Phosphorylation of another regulatory enzyme, GSK3β, in the liver was similarly altered. Moreover, the lower expression of GLUT4 and the reduced phosphorylation of PI3K in the soleus muscle of diabetic rats were upregulated by liraglutide. Interestingly, pretreatment with an opioid antagonist reversed these changes. These results clearly indicate that the repeat effect of liraglutide on glucose homeostasis is also dependent on endogenous BER.

However, the present study had some limitations. The present study is limited to only acute treatment with liraglutide. Moreover, extra-adrenal and/or pleiotropic effects might be observed during chronic treatment with liraglutide in rats with type-1 diabetes.

## 4. Materials and Methods

### 4.1. Materials

Liraglutide was obtained from Novo Nordisk (Bagsvaerd, Denmark). Exendin 9-39 (Ex9), naloxone, and other chemicals and reagents were purchased from Sigma-Aldrich (St. Louis, MO, USA).

### 4.2. Animal Model

Male Sprague–Dawley (SD) rats weighing 250 to 280 g were procured from the National Laboratory Animal Center (Taipei, Taiwan). After an overnight fast, rats received a single dose of STZ (65 mg/kg) by intravenous injection [27]. After seven days of STZ treatment, rats were considered to be diabetic if they had glucose concentration over 320 mg/dL. In the present study, the plasma glucose level before STZ injection was 99.41 ± 7.50 mg/dL, and after STZ injection was 366.32 ± 9.17 mg/dL.

Then, animal studies were completed within two weeks after diabetes onset. To minimize animal suffering, rats were anesthetized with an intraperitoneal injection of sodium pentobarbital (35 mg/kg) before the procedures. The experimental procedures performed in studies involving animals were approved (107061905) by the Institutional Animal Ethics Committee of Chi-Mei Medical Center and were in accordance with the NIH Guide for the Care and Use of Laboratory Animals.

### 4.3. Laboratory Determinations

Plasma glucose level was measured, following a previous method [27], by the glucose oxidase method using an analyzer (Quik-Lab, Ames; Miles Inc., Elkhart, IN, USA). BER levels were determined using a commercially available enzyme-linked immunosorbent assay (ELISA) kit (Peninsula Laboratories, Belmont, CA, USA).

### 4.4. Plasma BER Level in Rats with STZ-induced Diabetes

After fasting overnight, rats with STZ-induced diabetes received a subcutaneous injection of liraglutide. As rats receiving sodium pentobarbital have been demonstrated to have no significant change in plasma parameters [28], we used sodium pentobarbital (35 mg/kg i.p.) to anesthetize rats. The blood samples (0.1 mL) were collected from the tail vein for measuring plasma glucose and BER levels. In the preliminary study, liraglutide was found to produce the maximal plasma glucose-lowering effect in rats with STZ-induced diabetes 30 min after injection. Therefore, the effect of liraglutide on plasma BER was determined using blood samples collected at 30 min post-treatment. Rats with STZ-induced diabetes that received an injection of vehicle (0.9% saline) at the same volume as liraglutide were used as controls. Further experiments were performed using pretreatment with GLP-1R blocker, Ex9, the opioid receptor antagonist (naloxone), and others. The inhibitor was injected intravenously 30 min before liraglutide injection. In a continuous study to evaluate the chronic effect of liraglutide, all treatments were performed once daily for seven days, as described previously [21].

### 4.5. Adrenalectomy of Rats with STZ-induced Diabetes

Animals were anesthetized with sodium pentobarbital (35 mg/kg i.p.). Bilateral adrenalectomy was performed using the dorsal approach [21]. Prior to surgery, rats were fed ad libitum a standard t chow and 0.9% saline to drink. Rats in the sham operation (controls) were fed standard rat chow and tap water ad libitum. The animals appeared alert and in good health after surgery. Adrenalectomized rats received saline supplemented with corticosterone as a replacement for their drinking water for two weeks, which was then replaced by saline alone (gradually reduced within a week). After recovery, rats were then induced with type 1-like diabetes.

### 4.6. Isolation of Adrenal Medulla

At the end of the experiment, rats were sacrificed, the adrenal glands were quickly removed on ice, and the medulla were immediately dissected after removing the cortex. The tissues were cut into slices and incubated in ice-cold Krebs solution [29]. To understand the role of liraglutide in isolated adrenal tissues, the samples were pre-incubated with different concentrations of liraglutide. After 30 min, the medium was collected and frozen for further assay.

### 4.7. Quantitative Reverse-Transcription Polymerase Chain Reaction (qRT-PCR)

Total RNA from the adrenal medulla was isolated using TRIzol. The primers used in qRT-PCR were obtained from Roche (Roche Diagnostics GmbH, Germany). The concentration of each PCR product was calculated relative to a corresponding standard curve. Relative gene expression data were analyzed using real-time quantitative PCR and the 2-^ΔΔCq^ method. This gene expression was shown as the ratio of the target gene level to that of GAPDH, the housekeeping control, as described in our previous report [30]. Each sample was run in duplicate. The primers used were as follows: 5′-CCTATCGGGTGGAGCACTTC-3′ and 5′-TGGCTCTTCTCGGAGGTCAT-3′ (proopiomelanocortin [POMC]), 5′-CCAAGGTCATCCATGACGAC-3′ and 5′-TCCACAGTTCT GAGTGGC-3′ (GAPDH).

### 4.8. Western Blotting Analysis

After homogenizing, equal amounts of protein samples were separated using 10% sodium dodecyl sulfate-polyacrylamide gel electrophoresis. Following the transfer, membranes were blocked with 5% bovine serum albumin for 1 h and incubated with the following primary antibodies at 4 ˚C overnight: PEPCK (1:1000; cat. no. sc-32879, Santa Cruz Biotechnology, Dallas, TX, USA), GLUT-4 (1:1000; cat. no. sc-1608, Santa Cruz Biotechnology), GSK3β (1:1000; cat. no. sc-9166, Santa Cruz Biotechnology), phospho-GSK3β (Ser9) (1:1000; cat. no. sc-373800, Santa Cruz Biotechnology), PI3K (1:1000; cat. no. sc-423, Santa Cruz Biotechnology), phospho-PI3K (Tyr508) (1:1000; cat. no. sc-12929), β-tubulin (1:2000; cat. no. ab-6046, Abcam, Cambridge, LON, UK). On the next day, the membranes were incubated with secondary antibodies at room temperature for 1 h. Then, the blots were developed using a chemiluminescence kit. The densities of the bands were quantified using laser densitometry, as described in our previous report [31].

### 4.9. Statistical Analysis

The data were presented as means ± SEM. The data were analyzed by one-way ANOVA and Dunnett’s post-hoc test, or two-way ANOVA and Bonferroni’s post-hoc tests. The normal distribution of data was checked using the maximum likelihood method, with the significance level set at *p* < 0.05. Results were subjected to statistical analysis using SPSS 23 (Armonk, NY, USA).

## 5. Conclusions

The present study demonstrated that liraglutide activates GLP-1R to reduce hyperglycemia in rats with STZ-induced type-1 diabetes. This acute effect of liraglutide was mainly mediated by endogenous opioid BER through modifying glucose homeostasis.

## Figures and Tables

**Figure 1 pharmaceuticals-13-00407-f001:**
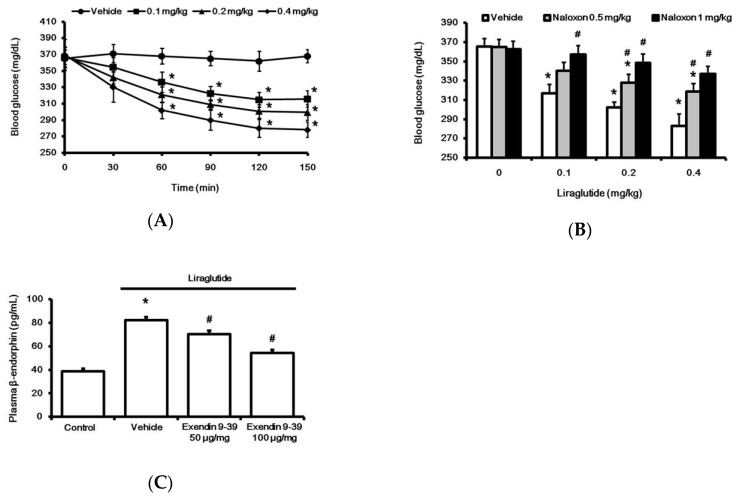
Effects of liraglutide on fasting plasma glucose and beta-endorphin (BER) levels in rats with tyep-1 diabetes. Acute injection of liraglutide attenuated plasma glucose in a time-dependent manner (**A**). The hypoglycemic effect of liraglutide was dose-dependently reversed by naloxone (**B**). Plasma glucose was obtained from rats that received liraglutide injection for 120 min. Plasma BER levels increased by liraglutide (0.4 mg/kg) were reversed in a dose-related manner by exendin 9-39 (Ex9), an established antagonist of the GLP-1 receptor (**C**). All results are shown as means ± standard errors (*n* = 6 per group). * *p* < 0.05 vs. vehicle-treated control. # *p* < 0.05 vs. liraglutide treated group.

**Figure 2 pharmaceuticals-13-00407-f002:**
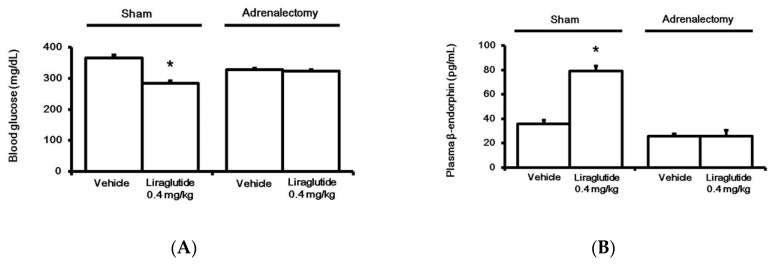
Effects of liraglutide on fasting plasma glucose (**A**) and beta-endorphin (BER) levels (**B**) in the adrenalectomy diabetic group or Sham group. Both groups were injected with liraglutide (0.4 mg/kg) for 120 min. Plasma glucose or BER levels changed by liraglutide were compared to the same volume in the vehicle-treated group (Veh). The results are shown as means ± standard errors (*n* = 6 per group). * *p* < 0.05 vs. the vehicle-treated control (Veh).

**Figure 3 pharmaceuticals-13-00407-f003:**
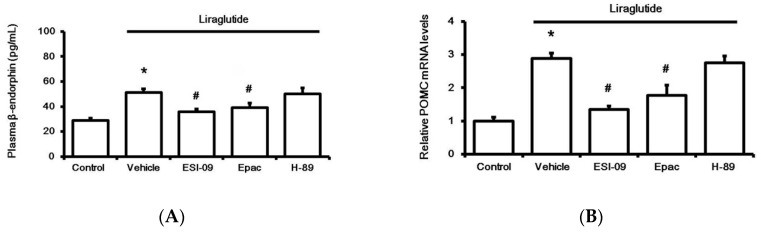
Effects of liraglutide (0.1 µM) on the adrenal medulla isolated from rats with type-1 diabetes. Influence of liraglutide was determined, after incubation for 1 h, via the changes in beta-endorphin (BER) secretion (**A**) and the mRNA levels of proopiomelanocortin (POMC) (**B**). Treatment with inhibitors, including 0.5 µM exendin 9-39 (Ex9) for GLP-1 receptor, ESI-09 (0.1 µM) for exchange protein directly activated by cAMP (Epac), and H-89 (0.5 µM) for protein kinase A (PKA), was administered 30 min before incubation with liraglutide. All results are shown as means ± standard errors (*n* = 6 per group). * *p* < 0.05 vs. control (Con). # *p* < 0.05 vs. vehicle-treated group (Veh).

**Figure 4 pharmaceuticals-13-00407-f004:**
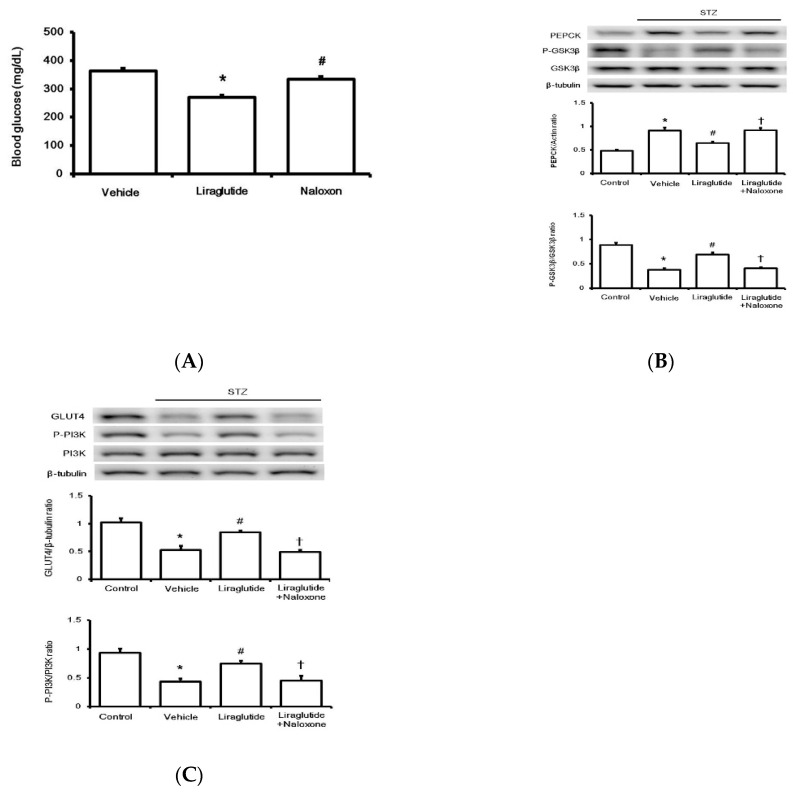
Effects of liraglutide on glucose regulation in rats with type-1 diabetes. The changes in glucose levels in diabetic rats after drug treatment (**A**). Glucose homeostasis was determined by Western blotting analysis for liver proteins PEPCK and GSK3β (**B**) and GLUT4 and PI3K phosphorylation in the soleus muscle (**C**) isolated from diabetic rats. Protein levels in diabetic rats (STZ) were compared with those in normal rats (Control), and diabetic rats receiving liraglutide injection once daily for seven days were compared with those receiving vehicle injection (Vehicle). Another group of diabetic rats receiving naloxone 30 min before liraglutide injection was also compared. Representative image of Western blots is shown in the upper figure, and quantification of the data is indicated in the lower figure. All results are shown as means ± standard errors (*n* = 6 per group). * *p* < 0.05 vs. control group. # *p* < 0.05 vs. vehicle group. † *p* < 0.05 vs. liraglutide treated group.

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
