# Peer review of "Liraglutide Activates Glucagon-Like Peptide 1 Receptor to Attenuate Hyperglycemia through Endogenous Beta-Endorphin in Diabetic Rats"

_pharmaceuticals, 2020, doi:10.3390/ph13110407_

Round 1

Reviewer 1 Report

The topic of the manuscript is of first importance. In my opinion, the study has been properly designed, but I have some doubts about the performing of the study (especially the number of rats in the examined groups) and statistical analysis. I recommend to consider the manuscript to be published after major revision done. Details are below:

  1. The number of rats in the examined groups seems to be too low (6 rats in every group) to have significant results and conclusions. Moreover, in the Material and methods, there is no information about the number of rats used in different part of the study.
  2. There is no information if any normality tests were used to check a normal distribution of data. It determines further statistical tests that can be used to assess the statistical significance of the results. If the study groups were so small (n=6), one could expect data was not normally distributed.
  3. In figure legends, all abbreviations should be explained independently of the main text. The study groups should also be described in figure legends (eg, what is the difference between control and vehicle treated group).
  4. Lines 141-143 - this paragraph was included by mistake to the manuscript, I suppose.
  5. In the Results section, there are some information which should be included only in the Material and methods section and there is no need to repeat them in the Results.

Author Response

The topic of the manuscript is of first importance. In my opinion, the study has been properly designed, but I have some doubts about the performing of the study (especially the number of rats in the examined groups) and statistical analysis. I recommend to consider the manuscript to be published after major revision done.

Reply: Thank you for your kind consideration of our submission. We are glad to have the opportunity to improve our manuscript as shown in red. Thank you very much.

Details of your helpful comments are below:

  1. The number of rats in the examined groups seems to be too low (6 rats in every group) to have significant results and conclusions. Moreover, in the Material and methods, there is no information about the number of rats used in different part of the study.

Reply: We appreciate your comments. The number of animals in each group is six. Each assay has been repeated for three times independently. Thank you very much.

  1. There is no information if any normality tests were used to check a normal distribution of data. It determines further statistical tests that can be used to assess the statistical significance of the results. If the study groups were so small (n=6), one could expect data was not normally distributed.

Reply: Thanks for the concerns. In the present study, the normal distribution to data has been checked using the maximum likelihood method.

  1. In figure legends, all abbreviations should be explained independently of the main text. The study groups should also be described in figure legends (eg, what is the difference between control and vehicle treated group).

Reply: Thank you for raising this useful indication. Following your instructions, we have amended the abbreviations and included the information of study groups in figure legends of this revision.

  1. Lines 141-143 - this paragraph was included by mistake to the manuscript, I suppose.

       Reply: Thank you for the helpful comments. We apologize for this oversight, and the text has been revised accordingly. Thank you very much.

  1. In the Results section, there are some information which should be included only in the Material and methods section and there is no need to repeat them in the Results.

      Reply: Thank you for the valuable comments. We have amended the manuscript accordingly. Thanking you again.

Reviewer 2 Report

In this study, Cheng et al. examined the effects of liraglutide, a GLP-1 receptor agonist, on endogenous opioid secretion in STZ rats. They found that acute treatment with liraglutide attenuated hyperglycemia in a dose-dependent manner and this effect was blocked by naloxone. Liraglutide treatment increased the plasma beta-endorphin (BER) level in diabetic rats and this effect was blocked by GLP-1 receptor antagonist. These effects were also reduced in adrenalectomized STZ rats. In the isolated adrenal medulla, liraglutide induced BER secretion and increased the BER mRNA level through Epac. Genes associated with gluconeogenesis in the liver were downregulated, whereas genes associated with glycolysis in muscle were upregulated by 7-day administration of liraglutide. Naloxone pretreatment inhibited these effects of liraglutide.

This study reveals the effects of liraglutide on endogenous beta-endorphin levels as the extra-pancreatic effects of GLP-1 analogs and is interesting. However, there are several critiques to be addressed.

  1. The title and abstract were somewhat too long and redundant. They should be more concisely described.

  1. Why the authors examined only beta-endorphin as endogenous opioid? The other endogenous opioid levels such as enkephalin or dynorphin were examined?

  1. Total adrenalectomy should induce adrenal insufficiency and might be lethal. The authors treated these rats with glucocorticoid supplementation? Both adrenal insufficiency and glucocorticoid supplementation should affect glucose metabolism and their findings. The authors should clarify this point.

  1. Introduction, line 58. It has been reported that GLP-1 receptor does not exist on alpha cell. This point should be clarified.

  1. Results, line 94. The time-dependent effect of liraglutide on blood glucose levels is important and should be more clearly presented as a figure.

  1. Change in blood glucose levels before and after STZ injection should also be described as a figure.

  1. Figure 1A was not a time-course change and should be revised to bar graph.

  1. The authors used STZ rats to evaluate insulin-independent effects of GLP-1, however, there remained endogenous insulin secretion in STZ rats. Therefore, it is difficult to consider “insulin-independent” effects in this model.

  1. Adrenalectomy blocked increase in plasma beta-endorphin levels with liraglutide treatment, but baseline plasma beta-endorphin levels appeared unchanged. It is difficult to assume that such a small change in beta-endorphin levels could have a major impact on glucose metabolism.

  1. Lines 124-130. The time-course and changes in blood glucose levels should be presented as a figure.

  1. It is unclear whether the changes in gluconeogenesis in the liver and glycolysis in the muscle was due to direct or indirect effects of liraglutide, as liraglutide ameliorated hyperglycemia. These changes might be secondary to change in blood glucose levels. Non-diabetic rats may be used to clarify this point.

  1. Discussion, line 184. Since the authors claims that the effects of liraglutide on plasma BER is through GLP-1 receptor, this sentence “However, studies…despite them lacking GLP-1R” did not make sense.

  1. Lines 141-143. The meaning of this sentence was not clear. Please look again the sentences carefully.

Author Response

Reply

ID: pharmaceuticals-989373

Title: Liraglutide activates glucagon-like peptide 1 receptor to attenuate hyperglycemia through endogenous beta-endorphin in diabetic rats

Reviewer 2:

In this study, Cheng et al. examined the effects of liraglutide, a GLP-1 receptor agonist, on endogenous opioid secretion in STZ rats. They found that acute treatment with liraglutide attenuated hyperglycemia in a dose-dependent manner and this effect was blocked by naloxone. Liraglutide treatment increased the plasma beta-endorphin (BER) level in diabetic rats and this effect was blocked by GLP-1 receptor antagonist. These effects were also reduced in adrenalectomized STZ rats. In the isolated adrenal medulla, liraglutide induced BER secretion and increased the BER mRNA level through Epac. Genes associated with gluconeogenesis in the liver were downregulated, whereas genes associated with glycolysis in muscle were upregulated by 7-day administration of liraglutide. Naloxone pretreatment inhibited these effects of liraglutide. This study reveals the effects of liraglutide on endogenous beta-endorphin levels as the extra-pancreatic effects of GLP-1 analogs and is interesting.

Reply: We appreciated your kind support and the helpful comments. Revision has been completed totally according to your suggestions. Please find our reply below.

There are several critiques to be addressed.

  1. The title and abstract were somewhat too long and redundant. They should be more concisely described.

Reply: Thank you for the helpful suggestions. The title and abstract section have been improved accordingly. Thank you very much.

  1. Why the authors examined only beta-endorphin as endogenous opioid? The other endogenous opioid levels such as enkephalin or dynorphin were examined?

Reply: Thank you for the insightful comments. Beta-endorphins are one of the naturally occurring opioid peptides. We agree that enkephalin or dynorphin should also be conducted. We are going to include this work in the future. Thank you very much.

  1. Total adrenalectomy should induce adrenal insufficiency and might be lethal. The authors treated these rats with glucocorticoid supplementation? Both adrenal insufficiency and glucocorticoid supplementation should affect glucose metabolism and their findings. The authors should clarify this point.

Reply: Thank you for raising this concern. Adrenalectomized rats received saline supplemented with corticosterone as a replacement for their drinking water for 2 weeks, which was then replaced by saline alone (gradually reduced within a week). We have included it in the revised version. Thank you very much.

  1. Introduction, line 58. It has been reported that GLP-1 receptor does not exist on alpha cell. This point should be clarified.

Reply: Thank you for highlighting this issue. We apologize for the oversight, and the errors in the references have been corrected according to your comments. Thank you very much.

  1. Results, line 94. The time-dependent effect of liraglutide on blood glucose levels is important and should be more clearly presented as a figure.

Reply: Thank you for the valuable comments. According to your suggestion, we have added the time-dependent effect of liraglutide on blood glucose levels in Figure 1A.

  1. Change in blood glucose levels before and after STZ injection should also be described as a figure.

Reply: Thank you for the thoughtful comments. We have added the blood glucose levels before and after STZ injection in Materials and Methods section.

  1. Figure 1A was not a time-course change and should be revised to bar graph.

Reply: Thank you for the helpful comments. Following your suggestion, the Figure 1A has been revised to bar graph.

  1. The authors used STZ rats to evaluate insulin-independent effects of GLP-1, however, there remained endogenous insulin secretion in STZ rats. Therefore, it is difficult to consider “insulin-independent” effects in this model.

Reply: Thank you for the valuable comments. We have improved the manuscript according to your suggestions. Thank you very much.

  1. Adrenalectomy blocked increase in plasma beta-endorphin levels with liraglutide treatment, but baseline plasma beta-endorphin levels appeared unchanged. It is difficult to assume that such a small change in beta-endorphin levels could have a major impact on glucose metabolism.

Reply: Thank you for highlighting this issue. In Figure 2B, the basal plasma beta-endorphin levels showed decreasing tendency in the adrenalectomy groups. Thank you very much.

  1. Lines 124-130. The time-course and changes in blood glucose levels should be presented as a figure.

Reply: Thank you for the valuable comment. The changes in blood glucose levels have been revised and presented in Figure 4A. Thank you very much.

  1. It is unclear whether the changes in gluconeogenesis in the liver and glycolysis in the muscle was due to direct or indirect effects of liraglutide, as liraglutide ameliorated hyperglycemia. These changes might be secondary to change in blood glucose levels. Non-diabetic rats may be used to clarify this point.

Reply: Thank you for the concerns. We totally agree with your comment that the changes of gluconeogenesis in the liver and glycolysis in the muscle should be evaluated. We will perform these related experiments in the future. Thank you very much.

  1. Discussion, line 184. Since the authors claims that the effects of liraglutide on plasma BER is through GLP-1 receptor, this sentence “However, studies…despite them lacking GLP-1R” did not make sense.

Reply: Thank you for the insightful comments. We have amended this sentence following your instructions.

  1. Lines 141-143. The meaning of this sentence was not clear. Please look again the sentences carefully.

Reply: Thank you for the helpful suggestion. We apologize for this oversight, and the text has been revised accordingly. Thank you very much.

Round 2

Reviewer 1 Report

The Authors have followed the reviewer's suggestions. Now, in my opinion, English in new parts of the manuscript should be corrected.

Author Response

Reply: Thank you very much. The language has been improved by the expert in editing service of MDPI. Hope you will like it. Thank you very much.

Reviewer 2 Report

The authors have revised the manuscript appropriately according to the reviewer's comments. 

Author Response

We appreciate your kind support. The language has been checked by the editing service of MDPI. Hope it will reach your standards. Thank you very much.
